# Facile Synthesis of 3-(Azol-1-yl)-1-adamantanecarboxylic Acids—New Bifunctional Angle-Shaped Building Blocks for Coordination Polymers

**DOI:** 10.3390/molecules24152717

**Published:** 2019-07-26

**Authors:** Dmitry Pavlov, Taisiya Sukhikh, Evgeny Filatov, Andrei Potapov

**Affiliations:** 1Kizhner Research Center, National Research Tomsk Polytechnic University, 30 Lenin Ave., 634050 Tomsk, Russia; 2Nikolaev Institute of Inorganic Chemistry, Siberian Branch of the Russian Academy of Sciences, 3 Lavrentiev Ave., 630090 Novosibirsk, Russia; 3Department of Natural Sciences, Novosibirsk State University, 2 Pirogov Str., 630090 Novosibirsk, Russia; 4Department of Chemical Technology, Polzunov Altai State Technical University, 46 Lenin Ave., 656038 Barnaul, Russia

**Keywords:** adamantane, carboxylate ligands, chain structures, heterocyclic ligands, metal-organic frameworks

## Abstract

For the first time, orthogonally substituted azole-carboxylate adamantane ligands were synthesized and used for preparation of coordination polymers. The angle-shaped ligands were prepared by the reaction of 1-adamantanecarboxylic acid and azoles (1*H*-1,2,4-triazole, 3-methyl-1*H*-1,2,4-triazole, 3,5-dimethyl-1*H*-1,2,4-triazole, 1*H*-tetrazole, 5-methyl-1*H*-tetrazole) in concentrated sulfuric acid. Variation of the solvent and substituents in azole rings allowed to prepare both 1D and 2D copper(II) and nickel(II) coordination polymers, [Cu_2_(trzadc)_4_(H_2_O)_0.7_]∙DMF∙0.3H_2_O, [Cu(trzadc)_2_(MeOH)]∙MeOH, [Ni(trzadc)_2_(MeOH)_2_] and [Cu_2_(mtrzadc)_3_(MeOH)]^+^NO_3_^–^ (trzadc-3-(1,2,4-triazol-1-yl)-adamantane-1-carboxylic acid; mtrzadc-3-(3-methyl-1,2,4-triazol-1-yl)-adamantane-1-carboxylic acid) which were structurally characterized by single crystal X-ray diffraction. Complex [Cu(trzadc)_2_(MeOH)]∙MeOH was shown to act as a catalyst in the Chan-Evans-Lam arylation reaction.

## 1. Introduction

Adamantane has been recognized for a long time as an important molecule for crystal design. An example of exploiting the potential of the adamantane moiety is 1,3,5,7-tetrakis(1,2,4-triazol-4-yl)adamantane [1], but even more simple 1,3,5,7-adamantanetetracarboxylic acid can produce spectacular frameworks [2]. A number of different adamantane derivatives have been designed and their coordination ability explored [3,4,5,6,7,8,9,10,11,12,13,14].

Metal-organic frameworks (MOFs) that feature metal ions with a ‘mixed’ coordination environment are often of great interest [15], and in case of adamantane usually achieved through a ‘mixed-ligand’ approach [16], which turns a two-component reaction into a much more complicated system, reducing reproducibility and ease of handling. One way to avoid this problem is to combine various functions in a single molecule, but no examples of coordination compounds built using two or more different donor groups linked by an adamantane backbone have been reported so far. This fact is probably due to synthetic difficulties in such functionalization [3,17]. Functional moieties with different coordination modes linked by a rigid platform like adamantane may be expected to form structures with complex geometry of great interest; producing a ‘mixed’ coordination environment for metal ion, so looking for ways to synthesize such species is always an actual challenge.

Following our interest in adamantane-based ligands [18], in this paper, we would like to communicate a method for the synthesis of the new adamantane-based ligands—3-(1,2,4-triazol-1-yl)-1-adamantanecarboxylic acid, 3-(tetrazol-1-yl)-1-adamantanecarboxylic acid and their methyl derivatives. Along with synthesis of new ligands themselves, several coordination polymers were synthesized and structurally characterized.

## 2. Results and Discussion

### 2.1. Symthetic Approach

For the synthesis of ligands, we employed the very well-known property of adamantane to form a carbocation with the charge localized at bridgehead carbon under highly acidic conditions [19,20,21] (Scheme 1).

This reaction went smoothly for both unsubstituted and 3,5-substituted 1,2,4-triazoles, as well as tetrazoles—much to our surprise, because usually, adamantylation of azoles can be achieved by substitution reaction of bromo- or hydroxy- groups [22,23,24,25,26]. Despite the fact that the carboxylic moiety destabilizes adamantyl cations to a significant extent, the reaction between azoles and 1-adamantanecarboxylic acid proceeded smoothly and gave new bifunctional adamantane derivatives in good yields. Hence, this reaction is an example of useful C-H functionalization.

To further explore interactions of 1-adamantylcarboxylic acid with different azoles, we attempted the same procedure for diazoles, but reactions with both pyrazole and imidazole were unsuccessful, and gave 3-hydroxyadamantane 1-carboxylic acid as products with 65% yield.

These observations correspond well with previous studies on the reactivity of azoles and adamantyl cations in acidic media [22,23,24,25,26], where it was determined that adamantylation with 1-adamantanol is possible with fairly acidic pyrazoles, the approximate limit to the pK_BH_+ being 0.8 [26]. Acidities of both unsubstituted pyrazole and imidazole are much lower (0.47 and 0.39) than that value [27].

### 2.2. Synthesis and IR Spectra of Coordination Polymers

Reaction of **trzadcH** with copper(II) and nickel(II) nitrates in methanol at 4:1 ratio afforded pale-blue and pale-green crystalline 1D coordination polymers **2** and **3**. Reaction with copper nitrate in DMF at the same ratio afforded bright-blue crystals **1** with different composition. At the same time, some amorphous green product precipitated. The reaction in dry DMF afforded only amorphous green solid, which turned blue on prolonged exposure to air. All attempts to obtain a crystalline form of this product have failed and no other attempts at characterization of the compound were made. Pure compound **1** could be prepared by addition of an equimolar amount of water. Reaction of **mtrzadcH** with copper(II) nitrate in methanol yielded bright-green crystals of 2D coordination polymer **4,** its higher dimensionality is probably due to the additional steric hindrance introduced by methyl group (see discussion below).

All of the IR spectra demonstrate coordination-induced shifts of bands corresponding to vibrations of azole rings and carboxylic groups (Appendix A).

The broad band at 3456 cm^−1^ in the spectrum of compound **1** can be attributed to lattice water. Bands of asymmetric and symmetric carboxylic C-O stretching vibrations appear at 1578 cm^−1^ and 1350 cm^−1^ (Δ 228 cm^−1^) compared to 1682 cm^−1^ and 1229 cm^−1^ (Δ 453 cm^−1^) in the uncoordinated ligand. For compounds **2** and **3** separation between carboxylic stretching vibration bands (Δ) is 265 and 185 cm^−1^, respectively. In case of compound **4**, Δ value is 314 cm^−1^. These observations are consistent with non-equivalence of metal-oxygen bonds in compounds 1–4 evident from structural data [28,29]. Stretching OH vibration bands of uncoordinated methanol can be seen in the spectrum of compound **2** at 3381 cm^−1^ but in the spectra of compounds **3** and **4** with coordinated methanol molecules these bands were not detected. Extremely intense band at 1415 cm^−1^ in the spectrum of compound **4** indicates the presence of uncoordinated NO_3_^−^ ion.

### 2.3. Crystal Structure of 3-(1,2,4-Triazol-1-yl)-1-adamantanecarboxylic Acid

Compound **trzadcH** crystallizes in a monoclinic crystal system, *P*2_1_/*n* space group, the unit cell contains four formula units of the compound (Figure 1). The molecules of **trzadcH** are involved in O–H···N intermolecular hydrogen bonding via carboxylic groups and nitrogen atoms in position 4 of 1,2,4-triazole cycles (O···N distance of 2.677 Å, O–H···N angle of 165.48°). Hydrogen bonds link the molecules into 1D chains, oriented at an angle of 39.2 degrees to the *c* axis (Appendix A).

In order to describe and compare the geometries of different angle-shaped N,O-ligands, we defined the angle ϕ_CctrN_ comprising the carbon atom of the carboxylic group, the centroid of the linker (adamantine in our case) and nitrogen donor atom (N atom at position 4 of triazole ring for compound **trzadcH**). The value of this angle for **trzadcH** (110.6°) is noticeably different from ϕ_CctrN_ values calculated for most of other angle-shaped ligands commonly used for the synthesis of coordination polymers (Table 1), indicating the perspectivity of the synthesized family of ligands for the construction of coordination polymers with new topologies.

### 2.4. Crystal Structures of Coordination Polymers

The non-isostructural complexes **1**–**3** are 1D coordination polymers revealing the bridging nature of the **trzadc^–^** ligand (Figure 2 and Figure 3). It coordinates to the metals via one N of triazole and one O atom of carboxyl moiety to form 1D chains. Two **trzadc^–^** are arranged between adjacent central atoms; thus, the ligands occupy four coordination sites of the metal forming a square environment. Other sites are occupied by water or methanol: two molecules in the complex **2**, revealing octahedral coordination environment of Ni, and one molecule in **3** with a square pyramidal arrangement of Cu. Compound **1** contains the first Cu atom in a square planar environment (without solvent molecules), while the second Cu atom features partially occupied (of 70%) water molecules in a coordination shell according to analysis of residual electron density map (Appendix A). Thus, this Cu atom has a mixed square pyramidal/square planar environment. The reason for partial occupancy of water can be corresponding with the bond distance Cu–O of 2.40 Å, being longer than typical Cu–OH_2_ bonds (of 1.95–2.20 Å) found in CSDB (v. 5.40, February 2019). As a result, the weak Cu–O interaction leads to defects in the crystal packing. Note that structure **1** contains partially occupied (of 30%) solvate water molecule close to coordinated one. However, they do not compete for the formation of hydrogen bonds (Appendix A), and there is enough space in the structure for their joint presence.

Overall geometry and crystal packing of 1D chains of complexes **1**–**3** are somewhat close (Appendix A). However, in **3**, adamantane units are arranged on one side from the CuNO plane (Appendix A), while in **1** and **2**, adamantane units lie on both sides of the corresponding plane. Furthermore, complex **3** reveals tight packing of the chains (with intramolecular hydrogen bonds between the methanol and carboxyl moiety), while **1** and **2** have solvate DMF, water or methanol molecules filling the space between the chains. The solvate molecules connect the chains via hydrogen bonds forming 2D-layered structures (Appendix A). 

The structure of complex **4** significantly differs from that of **1**–**3** because of the presence of a methyl substituent in the triazole moiety. This group prevents arrangement of the ligands to form a square coordination environment of Cu, which is observed for **1**–**3**. Instead, pentagonal pyramidal environment is implemented in **4** with two N atoms in apical and basal positions for the first Cu atom and one N atom in the apical position for the second one (Figure 3). The latter further coordinates the methanol molecule. In contrast to **1**–**3**, the carboxylate moiety serves as a bridging group connecting two adjacent central atoms. Thus, a 2D-layered structure of **4** is observed.

In principle, **trzadc^–^** and **mtrzadc^–^** ligands have two degrees of freedom, viz. rotation of triazole around C–N bond and rotation of carboxyl moiety around C–O bond (Appendix A). According to analysis of the corresponding torsion angles (Appendix A), conformation of the ligand can vary within certain limits, adjusting to the central atom environment.

### 2.5. Powder X-Ray Diffraction and Thermal Studies

Powder X-Ray diffraction was used to confirm phase purity of the bulk product. As experimental PXRD patterns match simulated ones sufficiently (Appendix A), we conclude that synthesis provides product of high purity with good crystallinity.

All of the coordination polymers demonstrate similar thermal stability. (Appendix A) In case of **1**, only DMF molecules are lost during the initial step (219–281 °C)—calc. mass loss 5.98%, observed 6.13%. Water loss along with the gradual organic ligand degradation started at 315 °C and completed at 430 °C. Compound **2** loses one uncoordinated methanol molecule during the first thermolysis step (90–200 °C), followed by deep degradation of the ligand and coordinated methanol loss in the range of 270–580 °C. As for **3**, coordinated methanol molecules remain bond up to the temperature of 190 °C, then a mass loss of 10.2% occurs, corresponding to two methanol molecules (calc. 10.4%). Coordination of polymer **4** demonstrated an extremely exothermic decomposition process in the range of 220–260 °C with a rapid mass loss, consistent with the presence of nitrate ions. 

### 2.6. Catalytic Activity Studies

Considering a vacant coordination site of a copper atom in compound **2** and coordinated solvent molecules in compounds **1** and **3**, as well as numerous applications of chain coordination polymers as catalysts [39,40], we decided to test their catalytic activity for the Chan-Evans-Lam N-arylation reaction (CEL arylation), which is known to be efficiently catalysed by various metal complexes [41] and copper salts [42].

We found that compound **2** serves as an efficient catalyst for the CEL arylation reaction, with imidazole and phenylboronic acid as model substrates (Scheme 2). The reaction was performed at 40° using methanol as a solvent on air with 1:1.5 molar ratio of reagents adding 5 mol% of the catalyst. With these conditions, we achieved 99% conversion (GC) of the starting imidazole after 24 h.

Interestingly enough, compound **1** demonstrated no catalytic activity in this reaction, as well as **3** and **4** (Table 2, entries 8–9), which allows us to conclude that not only copper itself is crucial, but also its coordination environment, being square planar in **2** and mixed square pyramidal/square planar in **1**. Also, it might be due to stronger bonding of water compared to the methanol molecules.

Furthermore, we found that methanol is an essential condition for the success of the reaction. We have tested a number of different solvents, some of them gave low conversions and in some cases, the polymer had dissolved completely (DMSO and water), in case of ethanol, partial dissolution was also noted; all results are summarized in Table 3. 

Additionally, we investigated the influence of reactant ratios and temperature. The reaction does occur at the room temperature, but it is inefficiently slow (Table 2, entry 2), giving only 60% conversion in 24 h. A decrease of the amount of boronic acid leads to the decrease of conversion (92%, Table 2, entry 3).

To elucidate the possibility of the homogenous catalysis happening and to prove our claim for the recyclability of the catalyst, we recovered the compound after running a reaction in a single cycle and run PXRD on the recovered sample (Figure 4). The sample was washed with methanol and dried on air prior to analysis.

As evident from the diffractogram, compound **2** remains crystalline and no new phases are observed, which proves that the process happening is heterogenous and our catalyst could be reused several times without losing its crystalline state.

## 3. Materials and Methods 

### 3.1. Materials

All reagents were of reagent grade and used as received without further purification. 3(5)-Methyl-1,2,4-triazole [43] and 3,5-dimethyl-1,2,4-triazole [44,45] were synthesized according to the literature. 

### 3.2. Methods

Thermogravimetric analysis (TGA) coupled to differential scanning calorimetry (DSC) was carried out with a NETZSCH 449F3 instrument (Erich NETZSCH GmbH & Co. Holding KG, Selb, Germany)) at a heating rate of 10 K/min in a stream of argon, scanning range—30–800 °C. Samples were air-dried for several days prior to analysis. GC-MS analyses were performed on an Agilent 7890A GC combined with an Agilent 5975C mass detector (Agilent Technologies, Santa Clara, CA, USA); carrier gas was helium. IR spectra were recorded on Agilent Cary 630 FTIR spectrometer equipped with a diamond ATR (attenuated total reflectance) tool. NMR spectra were recorded on a Bruker AVANCE III HD instrument (Bruker, Billerica, MA, USA) and are referenced to the solvent residual signal. Elemental analyses were carried out on Carlo Erba CHNS analyser (Val de Reuil, France).

#### 3.2.1. X-ray Structure Determination

Single-crystal X-ray diffraction data were collected at 150 K on a Bruker-DUO APEX CCD diffractometer (graphite monochromatized Mo Kα radiation, *λ* = 0.71073 Å, *φ* and *ω* scans of narrow frames, Bruker Corporation, Billerica, MA, USA) equipped with a 4K CCD area detector (Table 3). Absorption corrections were applied using the SADABS program [46]. The crystal structures were solved by direct methods and refined by full-matrix least-squares techniques with the use of the SHELXTL package [47] and Olex2 GUI [48]. Atomic thermal displacement parameters for non-hydrogen atoms were refined anisotropically. The positions of H atoms were calculated corresponding to their geometrical conditions and refined using the riding model. In the complex **1**, both coordinated and solvate water molecules show non-integer occupancies (of ca. 70/30% correspondingly) as clearly indicated by residual electron density map (Appendix A). Thus, the occupancies were refined with their sum constrained to unity to have reasonable atomic displacement parameters of O atoms.

#### 3.2.2. Ligand Synthesis

General method for preparation of all presented compounds is described below. Completion of the reactions was determined by TLC with bromocresol green solution as a stain.

10 mmol of the appropriate azole (1*H*-1,2,4-triazole, 3-methyl-1*H*-1,2,4-triazole, 3,5-dimethyl-1*H*-1,2,4-triazole, 1*H*-tetrazole or 5-methyl-1*H*-tetrazole) and 10 mmol of 1-adamantane carboxylic acid were placed in 20 mL screwcap vial and dissolved in 10 mL of 98% concentrated sulfuric acid. The mixture was cooled to 0 °C using an ice bath. After cooling, 10 mmol (1010 mg) of potassium nitrate were added in small portions over the period of 30 min. Once all of the nitrate has been added, the ice bath was removed, and the mixture was stirred for another 4 h at room temperature. After 4 h of stirring, reaction mixture was poured on 100 g of crushed ice with shaking and allowed to stand until it reached room temperature. Quenched mixture was filtered and neutralized with saturated NaHCO_3_ solution. Precipitate was filtered, washed with copious amounts of distilled water and dried in vacuum desiccator. Products are sufficiently pure for the synthesis of coordination compounds, but for the analysis the small portions of each compound were recrystallized from the mixture of water/MeOH (9:1). 

*3-(1,2,4-Triazol-1-yl)-adamantane-1-carboxylic acid***(trzadcH)** Yield 88%, colorless crystals. C_13_H_17_N_3_O_2_ (247.30): calcd. C 63.14, H 6.93, N 16.99; found C 63.38, H 7.09, N 17.13. Mp = 201–202 °C, ^1^H NMR (400 MHz, CDCl_3_): δ = 1.67 (s, 2 H, CH_2_), 1.82 (t, 2 H, CH), 2.08 (t, 4 H, CH_2_), 2.17 (s, 2 H, CH), 2.26 (s, 2 H, CH_2_), 7.97 (s, 1 H, trz), 8.59 (s, 1 H, trz), 12.53 (s, 1 H, COOH), ppm. ^13^C (100 MHz, CDCl_3_): δ = 29.0, 34.9, 37.7, 41.4, 42.2, 43.6, 58.5, 141.1, 151.3, 177.6, ppm. FT-IR (cm^−1^): ν = 2921 (s), 2849 (s), 1682 (s), 1506 (m), 1455 (w), 1363 (w), 1318 (w), 1285 (s), 1229 (s), 1132 (s), 1084 (s), 973 (m), 883 (w), 837 (s), 704 (s). *m/z* (methyl ester): 261 (100%), 202 (44%), 133 (91%), 100 (58%). 

Crystals suitable for single crystal X-Ray diffraction were obtainted by heating the compound in water to 150 °C in a sealed ampoule and slow cooling to room temperature. 

*3-(3-Methyl-1,2,4-triazol-1-yl)-adamantane-1-carboxylic acid***(mtrzadcH)** Yield 65%, colorless crystals. C_14_H_19_N_3_O_2_ (261.32): calcd. C 64.35, H 7.33, N 16.08; found C 64.62, H 7.07, N 16.41. Mp = 179–180 °C, ^1^H NMR (400 MHz, CDCl_3_): δ = 1.66 (s, 2 H, CH_2_), 1.80 (t, 2 H, CH), 2.04 (t, 4 H, CH_2_), 2.13 (s, 2 H, CH_2_), 2.23 (s, 5 H), 8.40 (s, 1 H, trz), 12.34 (s, 1 H, COOH), ppm. ^13^C (100 MHz, CDCl_3_): δ = 14.28, 29.0, 34.9, 37.7, 41.4, 42.2, 43.6, 58.10, 141.5, 159.6, 177.6, ppm. FT-IR (cm^−1^): ν = 3131 (w), 2922 (s), 2960 (s), 1685 (s), 1527 (m), 1458 (m), 1320 (m), 1244 (s), 1086 (m), 1023 (m), 867 (s), 717 (s). *m/z* (methyl ester): 275 (89%), 216 (28%), 193 (63%), 133 (100%), 91 (47%).

*3-(3,5-Dimethyl-1,2,4-triazol-1-yl)-adamantane-1-carboxylic acid***(dmtrzadcH)** Yield 47%, colourless needles. C_15_H_21_N_3_O_2_ (275.35): calcd. C 65.43, H 7.69, N 15.26; found C 65.71, H 7.48, N 15.49. Mp = 244–245 °C, ^1^H NMR (400 MHz, (CD_3_)_2_SO): δ = 1.49–1.79 (m, 7 H), 2.13 (m, 7 H), 2.23 (s, 4 H), 2.49 (s, 4 H), 12.25 (s, 1H, COOH), ppm. ^13^C (100 MHz, (CD_3_)_2_SO): δ = 14.0, 15.9, 29.3, 34.8, 37.6, 40.8, 42.3, 43.0, 60.1, 151.1, 156.6, 177.8, ppm. FT-IR (cm^−1^): ν = 3437 (w), 2915 (s), 2860 (s), 1691 (s), 1533 (m), 1408 (s), 1347 (m), 1257 (s), 1224 (s), 1078 (s), 989 (m), 891 (m), 851 (m), 739 (m), 693 (s). *m/z* (methyl ester): 289 (61%), 230 (25%), 193 (93%), 161 (48%), 133 (100%), 91 (40%).

*3-(Tetrazol-1-yl)-adamantane-1-carboxylic acid***(ttzadcH)** Yield 81%, white powder. C_12_H_16_N_4_O_2_ (248.28): calcd. C 58.05, H 6.50, N 22.57; found C 58.36, H 6.61, N 22.68. Mp = 195–197 °C, ^1^H NMR (400 MHz, (CD_3_)_2_SO): δ = 1.71 (s, 2 H), 1.85 (s, 4 H), 2.22 (s, 4 H), 2.32 (m, 4 H), 8.94 (s, 1 H, ttz), 12.34 (s, 1H, COOH), ppm. ^13^C (100 MHz, (CD_3_)_2_SO): δ = 29.06, 34.58, 37.49, 41.24, 42.25, 43.07, 64.09, 153.22, 177.33, ppm. FT-IR (cm^−1^): ν = 3145 (w), 2911 (s), 2862 (s), 1708 (s), 1458 (m), 1291 (m), 1260 (w), 1221 (s), 1165 (m), 1093 (m), 1092 (m), 1040 (s), 901 (m), 697 (s).

*1-(5-Methyl-tetrazol-1**-yl)-adamantane 3-carboxylic acid***(mttzadcH)** Yield 76%, white powder. C13H18N4O2 (262.31): calcd. C 59.53, H 6.92, N 21.36; found C 59.79, H 6.68, N 21.66. Mp = 179–180 °C, 1H NMR (400 MHz, (CD[3])_2_SO): δ = 1.88 (s, 2 H), 2.08 (m, 4 H), 2.41 (s, 4 H), 2.51 (s, 3 H), 2.57 (s, 2 H), 2.60 (s, 2 H), 11.14 (s, 1 H, COOH), ppm. 13C (100 MHz, (CD3)2SO): δ = 10.82, 28.99, 34.71, 37.30, 41.19, 42.59, 42.77, 63.46, 162.25, 181.68, ppm. FT-IR (cm^−1^): ν = 2932 (s), 2863 (s), 1708 (s), 1501 (m), 1458 (m), 1363 (w), 1336 (m), 1208 (s), 1142 (s), 1101 (m), 980 (w), 902 (m), 849 (m), 716 (s), 667 (s).

#### 3.2.3. Preparation of Coordination Compounds

All the complexes were prepared under solvothermal conditions as follows. 2 mL of 0.1 M ligand solution (0.2 mmol) in methanol (compounds **2**–**4**) or DMF (compound **1**) was mixed with 1 mL 0.1 M solution (0.1 mmol) of appropriate metal nitrate hydrate in a 4 mL screwcap vial. Vial was placed in oven at 80 °C for methanol and 120 °C for DMF for 24 h. Upon cooling to room temperature, crystals suitable for single crystal X-Ray analysis were formed, washed with 4 mL of fresh solvent and stored under pure solvent.

*[Cu_2_(trzadc)_4_(H_2_O)_0.7_]∙DMF∙0.3H_2_O***(1)** Bright blue crystals. Yield 23%. C_55_H_75_Cu_2_N_13_O_11_ (1221.35): calcd. C 54.07, H 6.19, N 14.91; found C 54.40, H 6.01, N 15.19. IR, cm^−1^: 3466 (v, O-H), 2907 (v, C-H), 1663 (v_a_, C-O), 1368 (v_s_, C-O). 

*[Cu(trzadc)_2_(MeOH)]∙MeOH***(2)** Pale blue crystals. Yield 40%. C_28_H_40_CuN_6_O_6_ (620.20): calcd. C 54.51, H 6.50, N 13.55; found C 54.24, H 6.37, N 13.80. IR, cm^−1^: 3338 (v, O-H), 2910 (v, C-H), 1611 (v_a_, C-O), 1346 (v_s_, C-O). 

*[Ni(trzadc)_2_(MeOH)_2_]***(3)** Pale green plates. Yield 32%. C_14_H_20_N_3_NiO_3_ (337.02): calcd. C 49.87, H 5.98, N 12.47; found C 49.63, H 6.21, N 12.42. IR, cm^−1^: 2895 (v, C-H), 1560 (v_a_, C-O), 1375 (v_s_, C-O). 

*[Cu_2_(mtrzadc)_3_(MeOH)]^+^NO_3_^–^***(4)** Bright green crystals. Yield 36%. C_43_H_58_CuN_10_O_10_ (938.53): calcd. C 55.01, H 6.23, N 14.93; found C 54.78, H 6.10, N 14.70. IR, cm^−1^: 2913 (v, C-H), 1606 (v_a_, C-O), 1415 (v, N-O), 1292 (v_s_, C-O). 

#### 3.2.4. Catalytic Activity Tests

In a typical procedure, 10 mL vial was loaded with 0.1 mmol (6.8 mg) of imidazole, 0.15 mmol (18.3 mg) of phenylboronic acid 1 mL of methanol and 6 mg of compound **2**. This mixture was left stirring on an oil bath at 40 °C for 24 h. Next, the mixture was centrifuged to recover the catalyst and supernatant analyzed by GC to calculate the conversion.

## 4. Conclusions

In summary, we have developed a pathway for the production of a new kind of adamantane-based ligands and successfully employed them for the preparation of several coordination polymers. The proposed method is robust, provides moderate to high yields and is easy to perform. It also allows a facile synthesis of the coordination polymers with mixed coordination environment, which may be very useful as demonstrated by catalytic activity of one of the complexes.

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
