# Peer review of "Facile Synthesis of 3-(Azol-1-yl)-1-adamantanecarboxylic Acids—New Bifunctional Angle-Shaped Building Blocks for Coordination Polymers"

_molecules, 2019, doi:10.3390/molecules24152717_

Round 1

Reviewer 1 Report

The manuscript submitted by Potapov and coworkers covers the synthesis and characterization of some interesting functionalized 1-adamantanecarboxylic acids. The introduction of orthogonal substituents in the ligands allows the use of them to built 1D and 2D coordination polymers with nice structures. Moreover, the paper is clear and easy to read. However, some point need to be modified in my opinion:

Line 16: Is 3-bromo-1-adamantane carboxylic acid used in the synthesis of the ligands? This is not consistent with Sheme 1 and section 3.2.2.

Abstract: The formulae of the coordination polymers 1-4 should be included and it should be specified that complex 2 is the one that displays catalytic activity.

Line 36: There are several examples of coordination compounds built with two carboxylate groups linked to the adamantane backbone. Do the authors mean two different donor groups? Or only two donor groups?

Line 70: Do the authors carried out any characterization on the amorphous green precipitate?

Line 80: Please, check the value provided for the difference between the asymmetric and symmetric carboxylic C-O stretching vibrations.  

Line 128: Does the tight packing of the chains of 3 gives crystals with a higher density than that of 1 and 2?

Section 3.2.2: The azole employed in each ligand synthesis should be included together with the technique employed to grow crystals of trzadcH suitable for single crystal X-ray diffraction.

Line 241: the name of dmtrzadcH is not complete.  

Once the above issues have been dealt with I would recommend publication.

Author Response

Line 16: Is 3-bromo-1-adamantane carboxylic acid used in the synthesis of the ligands? This is not consistent with Sheme 1 and section 3.2.2.

Typo corrected.

Abstract: The formulae of the coordination polymers 1-4 should be included and it should be specified that complex 2 is the one that displays catalytic activity.

Corrected.

Line 36: There are several examples of coordination compounds built with two carboxylate groups linked to the adamantane backbone. Do the authors mean two different donor groups? Or only two donor groups?

We mean two different donor groups, the phrase was corrected.

Line 70: Do the authors carried out any characterization on the amorphous green precipitate?

We did not attempt to perform a characterization of this compound – added to the text.

Line 80: Please, check the value provided for the difference between the asymmetric and symmetric carboxylic C-O stretching vibrations. 

Checked, typo corrected.

Line 128: Does the tight packing of the chains of 3 gives crystals with a higher density than that of 1 and 2?

Yes, as evident from Table 4 in the manuscript text, 3 is 0.034 g/cm3more dense than 1 and 0.026 g/cm3denser than 2.

Section 3.2.2: The azole employed in each ligand synthesis should be included together with the technique employed to grow crystals of trzadcH suitable for single crystal X-ray diffraction.

Corrected.

Line 241: the name of dmtrzadcH is not complete. 

Corrected.

Reviewer 2 Report

In this manuscript, convenient reactions of 1-adamantanecarboxylic acid and triazoles/tetrazoles are described, in addition to the preparation and crystal structures of four inorganic coordination compounds of 3-(tetrazol-1-yl)-1-adamantanecarboxylate complexes. Direct reactions between unsubstituted 1-adamantanecarboxyilc acid and azoles are great findings for not only ligand syntheses but also general organic chemistry. Although syntheses and crystal structures of inorganic complexes are unfortunately in the range of routine work, it is acceptable as a demonstration of newly prepared bridging ligand trzadcH. I recommend to publish this manuscript in Molecules after revising following points.

1) Please append experimental detail of catalytic activity tests.

2) In line 158, a small peak at 329 C in the DSC diagram (fig. S8) is quite suspicious for phase transition. It should be confirmed by spectroscopic method. It seems not important for the characterization and I recommend to remove it.

3) For crystal structure of 1, non-integer occupancy for crystal solvent atom is often the case. On the other hand, the 70 % occupancy for the coordinating oxygen atom to the Cu2+ ion is unnatural (should be 100 %). If such a disordering is present, whole the coordination environment must be disordered too. As the cif is not attached and I cannot do further inspection, please confirm it again.

4) In line 102, what kind of new structure is expected by the bridging angle of trzadcH?

Author Response

1) Please append experimental detail of catalytic activity tests.

Corrected.

2) In line 158, a small peak at 329 C in the DSC diagram (fig. S8) is quite suspicious for phase transition. It should be confirmed by spectroscopic method. It seems not important for the characterization and I recommend to remove it.

The sentence was removed.

3) For crystal structure of 1, non-integer occupancy for crystal solvent atom is often the case. On the other hand, the 70 % occupancy for the coordinating oxygen atom to the Cu2+ ion is unnatural (should be 100 %). If such a disordering is present, whole the coordination environment must be disordered too. As the cif is not attached and I cannot do further inspection, please confirm it again.

This is indeed an atypical case of disorder of coordinated water molecule with the rest ordered coordination environment of Cu, which was revealed by an analysis of residual electron density map as shown in Fig. S14 (supplementary information). The reference to Fig. S14 was accidentally missed, and we apologize for that. In fact, both coordination polyhedra for Cu with and without water molecule (square pyramidal and square planar) are reasonable and have analogues among other polyhedra in the compounds 1 and 3.

CIF files are added as supplementary materials to allow the inspection.

4) In line 102, what kind of new structure is expected by the bridging angle of trzadcH?

As it is impossible to 100% accurately predict the resulting structures of coordination compounds, we are not meaning any particular type of structure. Instead, it is a general assumption, which seems logical as the trzadcH fills the gap in the bridging angles range.

Reviewer 3 Report

The paper “Facile synthesis of 3-(azol-1-yl)-1-adamantanecarboxylic acids – new bifunctional angle-shaped building blocks for coordination polymers” describes synthesis of five new bifunctional ligands, their use for the preparation of coordination polymers, and the application in heterogeneous catalysis.

The synthetic procedures are well described with all characterization data included in the SI. In my opinion, a brief discussion of the powder XRD patterns of the bulk coordination polymers would improve the clarity.

However, I see the catalytical part as the weakest point of the manuscript. To this part I have following comments:

1.      The authors have to include the experimental details of the experiments in chapter 3.2 Methods.

2.      In chapter  2.6 Catalytic activity studies are discussed catalytic results that are not included in Table 3. The authors should include all experiments in the Table 3 and addition of the catalytic conditions would improve the clarity of the paper.

3.      In table 3, it is stated that in DMSO and water the coordination polymer degraded. Was the polymer dissolved in these solvents? If yes, was also in other solvents observed partial dissolution?

To summarize, I recommend the paper for publishing in Molecules after addressing the above mentioned comments.

Author Response

1.      The authors have to include the experimental details of the experiments in chapter 3.2 Methods

Details of the catalysis experiment were added as section 3.2.4

2.      In chapter  2.6 Catalytic activity studies are discussed catalytic results that are not included in Table 3. The authors should include all experiments in the Table 3 and addition of the catalytic conditions would improve the clarity of the paper.

All discussed experiments have been gathered in the table. Reaction conditions were added to the footer.

3.      In table 3, it is stated that in DMSO and water the coordination polymer degraded. Was the polymer dissolved in these solvents? If yes, was also in other solvents observed partial dissolution? 

Yes, the polymer was completely dissolved in these solvents. Partial dissolution was also observed in EtOH, presumably due to the presence of 4% of water in the alcohol. – added to the text

Reviewer 4 Report

Potapov et al synthesised five new hybrid organic ligands aiming to study their coordination abilities, however only two of them were used to obtain the coordination compounds, therefore the need of reporting the synthesis and characterisation of the remaining two (unused) ligands is questioned. Moreover, the study is not systematic in terms of the use of the second ligand, since only one compound is obtained, against three with the first ligand, therefore authors should either remove the compound with the second ligand or either add more data. Also, the catalytic studies should be extended to a) the metal salts, for comparison, b) catalyst loading, for understanding purposes, and c) recoverability experiments (with PXRD if possible), to elucidate the homo- or heterogeneous nature of the catalytic procedure.  In terms of references and especially for reviews with 1D coordination polymers, a couple of reviews Chem. Rev., 2011, 111, 688 & Coord. Chem., 2018, 71, 371, may have not attracted the interest of the team. Other than that, the study is of high standards, fulfils the publication criteria of the journal and will attract the interest of the research community, therefore should be accepted for publication after revision.

Author Response

Potapov et al synthesised five new hybrid organic ligands aiming to study their coordination abilities, however only two of them were used to obtain the coordination compounds, therefore the need of reporting the synthesis and characterisation of the remaining two (unused) ligands is questioned. Moreover, the study is not systematic in terms of the use of the second ligand, since only one compound is obtained, against three with the first ligand, therefore authors should either remove the compound with the second ligand or either add more data.

We admit that this study might not seem very systematic in the terms of use of all presented ligands, however, as the name of the manuscript suggests, we aimed to demonstrate a new family of the adamantane-based ligands as the main goal and coordination compounds with their properties are there to support our claims. Hence, we wish to keep all the ligands in the text.

Also, the catalytic studies should be extended to a) the metal salts, for comparison, b) catalyst loading, for understanding purposes, and c) recoverability experiments (with PXRD if possible), to elucidate the homo- or heterogeneous nature of the catalytic procedure. 

As for the catalytic studies, we do not think that extending the study to metal salts is necessary, because the main reason to use our compound as catalyst is the heterogeneous nature of the reaction. Moreover, CEL reaction is relatively well-known to be catalyzed by copper salts, such as copper acetate. We added a reference to the text for the reader to have the opportunity to compare two procedures. We agree with the need for recoverability experiments, so we included PXRD pattern of the recovered catalyst after a single cycle (Figure 4). Considering the catalyst loading, we decided to go with 5 mol% because of the relatively low activity of the catalyst, and decrease in its amount is undesirable. Increase in the amount of the catalyst wasn’t investigated because the yield of coordination polymer is not that high and a slight increase in the rate of the reaction doesn’t justify this

In terms of references and especially for reviews with 1D coordination polymers, a couple of reviews Chem. Rev.2011111, 688 & Coord. Chem.201871, 371, may have not attracted the interest of the team. Other than that, the study is of high standards, fulfils the publication criteria of the journal and will attract the interest of the research community, therefore should be accepted for publication after revision.

We found suggested references perfectly suitable and useful to the discussion and added them to the manuscript. 

Round 2

Reviewer 3 Report

The authors have addressed all points and for this reason I am happy to recomend the paper for publishing.

Reviewer 4 Report

Authors did not modify their text as suggested, however by adding new references for comparison and performing PXRD studies to prove catalyst recovering and stability, they have significantly improved this work and therefore is now suitable for publication.